# Quantum capacities of transducers

Chiao-Hsuan Wang [1,2,3,4] ✉, Fangxin Li[4] & Liang Jiang [4]

High-performance quantum transducers, which faithfully convert quantum information between disparate physical carriers, are essential in quantum science and technology. Different figures of merit, including efficiency, bandwidth, and added noise, are typically used to characterize the transducers' ability to transfer quantum information. Here we utilize quantum capacity, the highest achievable qubit communication rate through a channel, to define a single metric that unifies various criteria of a desirable transducer. Using the continuous-time quantum capacities of bosonic pure-loss channels as benchmarks, we investigate the optimal designs of generic quantum transduction schemes implemented by transmitting external signals through a coupled bosonic chain. With physical constraints on the maximal coupling rate $g_{max}$, the highest continuous-time quantum capacity $Q^{max} \approx 5g_{max}$ is achieved by transducers with a maximally flat conversion frequency response, analogous to Butterworth electric filters. We further investigate the effect of thermal noise on the performance of transducers.

Classically, transducers are devices, such as antenna and microphones, that can convert signal from one physical platform to another. In quantum technology, transducers are essential elements that can faithfully convert quantum information between physical systems with disparate information carriers[1–3]. High-performance quantum transducers are the key to realize quantum networks[4–7] by interconnecting local quantum processors, such as microwave superconducting systems[8,9], with long-range quantum communication carriers, such as optical fibers[10]. Tremendous progress has been made in a variety of coherent platforms for microwave-to-optical[11–23], microwave-to-microwave[24,25], and optical-to-optical[26–29] frequency conversion.

Coherent conversion of quantum information between distinct devices is a challenging task. A functional quantum transducer has to satisfy demanding criteria simultaneously—high conversion efficiency, broad bandwidth, and low added noise—and its performance has been characterized by these three figures of merit[30]. On the other hand, a unified metric to assess the quantum communication capability of transducers is lacking. For example, one transducer may have a high conversion efficiency but operates within a narrow bandwidth, another may allow broadband conversion at a lower efficiency. It is hard to compare their transmission capability given separate criteria.

Quantum capacity, the highest achievable quantum communication rate through a channel[31–34], provides a natural metric to characterize the performance of quantum transducers. Consider a generic direct quantum transduction process by propagating external signals through a coupled bosonic chain[35]. After sending an input signal through the transducer, the output signal will be a mixture of the input signal and environmental noise. Assuming the environmental noise is thermal and that the transducer has no amplification effect, the action of the transducer can be described as a bosonic thermal-loss channel that attenuates the input state and combines it with a noisy thermal state[36]. We can thus model direct quantum transducers as bosonic thermal-loss channels and evaluate their quantum capacities.

In this article, we use quantum capacity to assess the intrinsic quantum communication capability of transducers. Using the continuous-time pure-loss quantum capacities of transducers as benchmarks, we discover that the optimal designs of transducers are those with maximally flat frequency response around the unity-efficiency conversion peak. Under the physical constraint of a bounded maximal coupling rate $g_{max}$ between the bosonic modes, the maximal continuous-time quantum capacity $Q^{max} \approx 5g_{max}$ is achieved by maximally flat transducers implemented by a long bosonic chain. We further include the effect of thermal noise from the environment by considering additive lower and upper bounds on quantum capacities of thermal-loss channels. Our methods provide a unified quantity

[1]Department of Physics and Center for Theoretical Physics, National Taiwan University, Taipei 10617, Taiwan. [2]Center for Quantum Science and Engineering, National Taiwan University, Taipei 10617, Taiwan. [3]Physics Division, National Center for Theoretical Sciences, Taipei 10617, Taiwan. [4]Pritzker School of Molecular Engineering, University of Chicago, Chicago, IL 60637, USA. ✉e-mail: chiaowang@phys.ntu.edu.tw

to assess the performance of transducers across various physical platforms and suggest a fundamental limit on the quantum communication rate set by the physical coupling strength.

## Results

### Capacity as a metric for quantum transducers

We use the concept of quantum capacities of bosonic channels to assess the performance of direct quantum transducers. The quantum capacity quantifies the maximal achievable qubit communication rate through a quantum channel. Here we focus on direct quantum transduction achieved by directly converting quantum signals between bosonic modes via a coherent interface. At a given frequency $\omega$ in the appropriate rotating frame, assuming no intrinsic losses and no amplification gain, a direct quantum transducer with conversion efficiency $\eta[\omega]$ can be modeled as a Gaussian thermal-loss channel[36] described by the relation between the input and output modes, up to phase shifts,

$$\hat{b}_{\text{out}}[\omega] = \sqrt{\eta[\omega]}\,\hat{a}_{\text{in}}[\omega] - \sqrt{1-\eta[\omega]}\,\hat{b}_{\text{in}}[\omega], \qquad (1)$$

where $\hat{a}_{\text{in}}[\omega]$ is the input signal mode sent out by Alice, $\hat{b}_{\text{out}}[\omega]$ is the output signal mode received by Bob, and $\hat{b}_{\text{in}}[\omega]$ is the noisy input state from the environment with a mean thermal photon number $\bar{n}[\omega] = \langle \hat{b}_{\text{in}}^{\dagger}[\omega]\hat{b}_{\text{in}}[\omega] \rangle$ (see Fig. 1a). Note that we have no access to the reflective signal at Alice's side.

When the thermal photon number from the environment is negligible, $\bar{n} \approx 0$ for optical systems or via cooling[25,37], this special case of thermal-loss channels is called the pure-loss channel. For pure-loss channels, their capacities are additive and can be analytically determined. Specifically, for one-way quantum communication (for example, from Alice to Bob only), for discrete-time signals at a given frequency $\omega$ with a fixed conversion efficiency $\eta[\omega]$, the one-way pure-loss capacity is given by[38]

$$q_1[\omega] = \max\left\{\log_2\left(\frac{\eta[\omega]}{1-\eta[\omega]}\right), 0\right\}, \qquad (2)$$

which is the maximal amount of quantum information that can be reliably transmitted per channel use. This channel has infinite quantum capacity for ideal conversions, $\eta \to 1$, $q_1 \to \infty$, and has vanishing capacity when more than half of the signal is lost, $\eta \in [0, 1/2)$, $q_1 = 0$.

In reality, a quantum transducer has a finite conversion band and the conversion efficiency should be frequency-dependent. Treating different frequency modes within the conversion band as

parallel quantum channels and taking the continuous limit in $\omega$, here we define a continuous-time one-way pure-loss capacity of a quantum transducer,

$$Q_1 \equiv \int q_1[\omega]\,d\omega/2\pi. \qquad (3)$$

In contrast to the discrete-time one-way pure-loss capacity expression Eq. (2) that quantifies the maximal achievable quantum communication rate per channel use, the continuous-time quantum capacity defined in Eq. (3) is the maximal amount of quantum information that can be reliably transmitted through the transducer per unit time. This form of capacity is a direct analog to the Shannon capacity of classical continuous-time communication channels subject to frequency-dependent uncorrelated noises[39].

If the pure-loss channel is further assisted by two-way classical communication (between Alice and Bob) and local operations, the corresponding discrete-time two-way pure-loss capacity[40] is given by

$$q_2[\omega] = -\log_2(1 - \eta[\omega]). \qquad (4)$$

This channel again has infinite quantum capacity for ideal conversions, $\eta \to 1$, $q_2 \to \infty$, but has vanishing capacity only when the efficiency goes to zero, $\eta \to 0$, $q_2 = 0$. The corresponding continuous-time two-way pure-loss capacity is defined as

$$Q_2 \equiv \int q_2[\omega]\,d\omega/2\pi. \qquad (5)$$

The continuous-time pure-loss quantum capacities $Q_1$ and $Q_2$ defined above incorporate both concepts of efficiency and bandwidth and set the fundamental limit on the quantum communication rate based upon intrinsic transducer properties. To characterize these maximal achievable rates, we have assumed that infinite energy is available at the transducers. In practice, quantum capacities of transducers shall be lower in energy-constrained scenarios[41,42]. We emphasize that $Q_1$ and $Q_2$ have the unit of qubits per second, and we will show in later text that these highest achievable communication rates are linked to the maximal coupling rates in the underling physical transducer system.

### Physical limit on the quantum capacities of transducers

The conversion efficiency of a transducer, $\eta[\omega]$, is determined by the parameters of its underlying physical implementation. We are

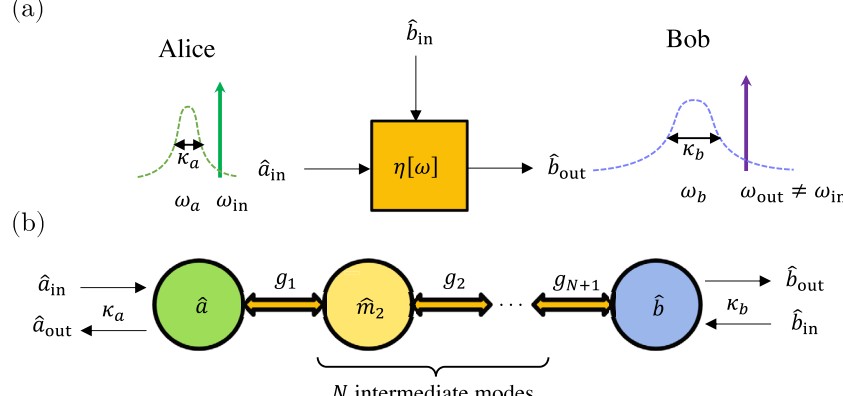

**Fig. 1 | Generic model of quantum transducers. a** A quantum transducer that can faithfully convert quantum states between different input and output frequencies $\omega_{\text{in}}$ and $\omega_{\text{out}}$ (in the lab frame), which is modeled as a thermal-loss channel with transmittance $\eta[\omega]$. **b** Schematic of a $N$-stage quantum transducer through a coupled bosonic chain connected to external input and output signals.

interested in how the quantum capacities of transducers $Q_1$ and $Q_2$ are limited by the physical parameters of the transduction platform. Consider the generic model of direct quantum transducer[11–25,27] implemented by a coupled bosonic chain with $N+2$ bosonic modes $\hat{m}_j$, where the two end modes, $\hat{m}_1 = \hat{a}$ and $\hat{m}_{N+2} = \hat{b}$, are coupled to external signal input and output ports at rates $\kappa_1 = \kappa_a$ and $\kappa_{N+2} = \kappa_b$, respectively (see Fig. 1b). Coherent quantum conversion can be realized by propagating bosonic signals from mode $\hat{a}$ (at frequency $\omega_a$) to mode $\hat{b}$ (at frequency $\omega_b$) through $N$ intermediate stages, and we call this interface a $N$-stage quantum transducer. The conversion efficiency of a $N$-stage transducer is a frequency-dependent function determined by system parameters[12,35],

$$\eta_N = \eta_N[\omega](\kappa_a, \kappa_b, \{\Delta_j\}, \{g_j\}), \tag{6}$$

where $\Delta_j$ is the detuning of mode $\hat{m}_j$ in the rotating frame of the laser drive(s) that bridges the up- and down-conversions between the input and output signals, and $g_j$ is the coupling strength of the beam-splitter type interaction between the neighboring bosonic pair $\hat{m}_j$ and $\hat{m}_{j+1}$. Here we have assumed the system has no intrinsic losses and we will take $g_j$'s to be real and positive without loss of generality.

For realistic physical implementations, the coherent coupling between neighboring modes is typically the most demanding resource. Therefore, under the physical constraint $\forall j, g_j \leq g_{\max}$, we look for the optimized choice of parameters $\kappa_a$, $\kappa_b$, $\Delta_j$'s, and $g_j$'s to achieve the maximal possible $Q_1$ and $Q_2$ for $N$-stage quantum transducers. To attain the highest possible capacity, the physical parameters of the transducer have to satisfy the generalized matching condition[35] such that $\eta_N[\omega_c] = 1$ at some frequency $\omega_c$. Note that the physics of the system is invariant under an overall shift in energy by choosing a different rotating frame, which corresponds to the relocation of $\omega_c$.

Using the continuous-time pure-loss capacities as the benchmarks, we find that maximal values of $Q_1$ and $Q_2$ are achieved when the $N$-stage quantum transducer has a maximally flat (MF) efficiency,

$$\left.\frac{\partial \eta_N^{\mathrm{MF}}[\omega]}{\partial \omega}\right|_{\omega = \omega_c} = \cdots = \left.\frac{\partial^{2N+3} \eta_N^{\mathrm{MF}}[\omega]}{\partial \omega^{2N+3}}\right|_{\omega = \omega_c} = 0. \tag{7}$$

Intuitively, with a flat plateau around $\eta_N[\omega_c] = 1$, this maximally flat transducer design guarantees a local maximum for $Q_1$ and $Q_2$, and we have seen strong numerical evidence that this solution is likely a global maximum as well under the physical constraint $\forall j, g_j \leq g_{\max}$ (see Methods). In the later discussion, we will use this as an optimized design for $N$-stage transducers. For $N$-stage transducers under the above physical constraint, we find that the optimal parameters satisfying Eq. (7), denoted by $\star$, are

$$\kappa_a^\star = \kappa_b^\star = 2\sqrt{\frac{\sin\left[\frac{3\pi}{2(N+2)}\right]}{\sin\left[\frac{\pi}{2(N+2)}\right]}} g_{\max}, \tag{8}$$

$$g_j^\star = \sqrt{\frac{\sin\left[\frac{\pi}{2(N+2)}\right] \sin\left[\frac{3\pi}{2(N+2)}\right]}{\sin\left[\frac{(2j-1)\pi}{2(N+2)}\right] \sin\left[\frac{(2j+1)\pi}{2(N+2)}\right]}} g_{\max}, \tag{9}$$

and $\forall j, \Delta_j^\star = -\omega_c$ (see Methods). Note that the optimized parameters are symmetric, $g_j^\star = g_{N+2-j}^\star$, $\kappa_a^\star = \kappa_b^\star$, and $g_1^\star = g_{N+1}^\star = g_{\max}$.

A $N$-stage maximally flat transducer is a direct analog to a $(N+2)$th order Butterworth low-pass electric filter (see Methods). The

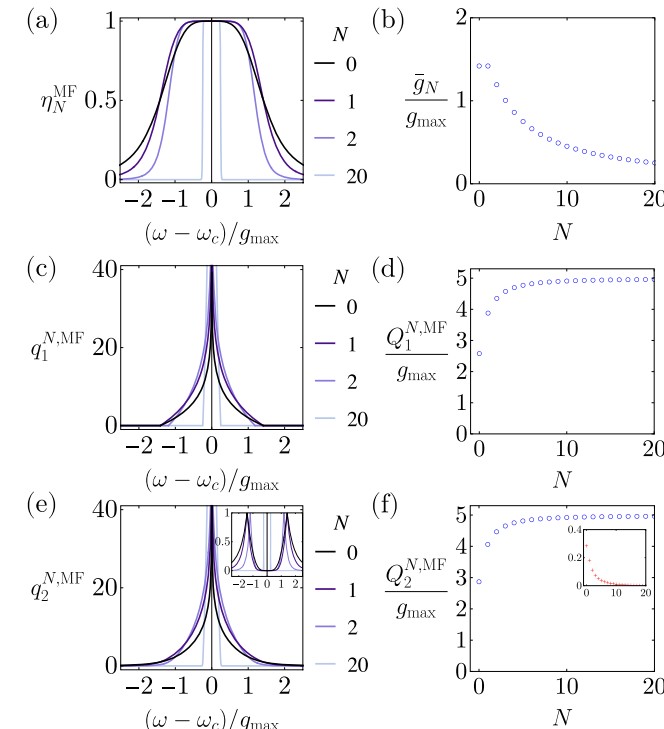

**Fig. 2 | Diagrams for $N$-stage quantum transducers with maximally flat conversion efficiency. a** Maximally flat efficiency function $\eta_N^{\mathrm{MF}}[\omega]$ for different $N$. **b** The mean coupling $\bar{g}_N$ as a function of $N$. **c** The discrete-time one-way pure-loss capacity, $q_1^{N,\mathrm{MF}}[\omega]$, for different $N$. **d** The continuous-time one-way pure-loss capacity, $Q_1^{N,\mathrm{MF}}$, as a function of $N$. **e** The discrete-time two-way pure-loss capacity, $q_2^{N,\mathrm{MF}}[\omega]$, for different $N$. Inset shows the gain in capacity assisted by the two-way protocol, $q_2^{N,\mathrm{MF}}[\omega] - q_1^{N,\mathrm{MF}}[\omega]$. **f** The continuous-time two-way pure-loss capacity, $Q_2^{N,\mathrm{MF}}$, as a function of $N$. Inset shows the gain in capacity assisted by the two-way protocol, $(Q_2^{N,\mathrm{MF}} - Q_1^{N,\mathrm{MF}})/g_{\max}$.

maximally flat efficiency $\eta_N^{\mathrm{MF}}[\omega]$ has a general form

$$\eta_N^{\mathrm{MF}}[\omega] = \frac{1}{((\omega - \omega_c)/\bar{g}_N)^{2(N+2)} + 1}, \tag{10}$$

where

$$\bar{g}_N \equiv 2\sqrt{\sin\left[\frac{\pi}{2(N+2)}\right] \sin\left[\frac{3\pi}{2(N+2)}\right]} g_{\max}. \tag{11}$$

Here $\bar{g}_N$ is the mean coupling given by $\bar{g}_N = \sqrt[N+2]{\sqrt{\kappa_a^\star \kappa_b^\star} \prod_{j=1}^{N+1} g_j^\star}$, which can be inferred from Eq. (23) in Methods. $\bar{g}_N$ also has the physical meaning of the transducer bandwidth—the full width at half maximum of $\eta_N^{\mathrm{MF}}[\omega]$ is $2\bar{g}_N$. The value of $\bar{g}_N/g_{\max}$ monotonically decreases with $N$ as shown in Fig. 2b. The monotonically decreasing $\bar{g}_N$ might seem counter-intuitive at first glance, but the choice of parameters actually enables maximally flat transmission band, which can take the full advantage of the diverging channel capacity at $\eta[\omega_c] = 1$ to optimize the overall performance under the given physical constraint.

Given this general form, we can find their discrete-time pure-loss capacities at a given frequency, $q_1^{N,\mathrm{MF}}[\omega]$ and $q_2^{N,\mathrm{MF}}[\omega]$, and then evaluate the continuous-time pure-loss capacities of the maximally flat transducers (see Fig. 2c–f). Specifically,

$$Q_1^{N,\mathrm{MF}} = \frac{2(N+2)}{\pi \log(2)} \bar{g}_N, \tag{12}$$

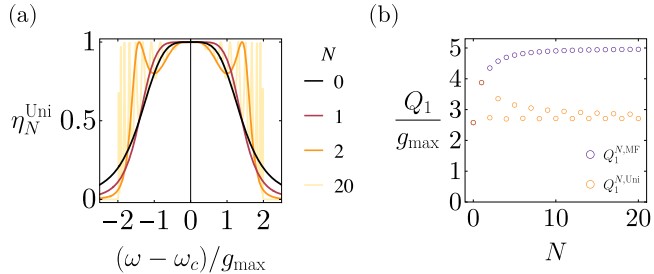

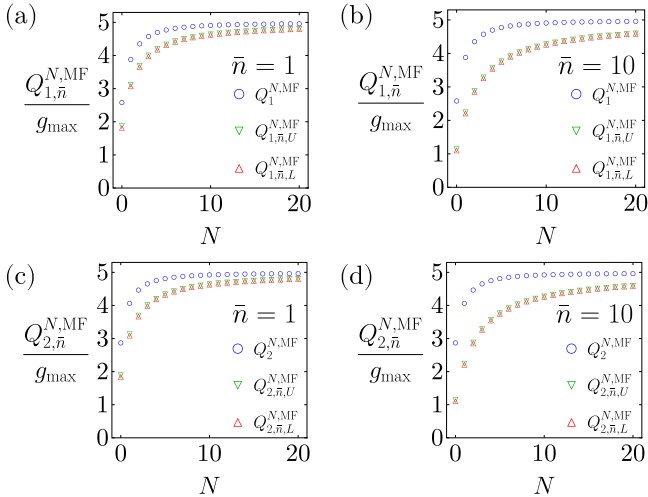

**Fig. 3 | Diagrams for N-stage quantum transducers with uniform couplings.**
**a** Optimal efficiency function $\eta_N^{\mathrm{Uni}}[\omega]$ for N-stage transducers with uniform couplings. **b** Continuous-time one-way pure-loss quantum capacities of N-stage maximally flat transducers $Q_1^{N,\mathrm{MF}}$ (purple) and uniform transducers $Q_1^{N,\mathrm{Uni}}$ (orange).

**Fig. 4 | Quantum capacities of maximally flat transducers under thermal loss.**
**a** Continuous-time one-way thermal-loss capacity upper and lower bounds with mean thermal photon number $\bar{n}=1$. **b** Continuous-time one-way thermal-loss capacity upper and lower bounds with mean thermal photon number $\bar{n}=10$.
**c** Continuous-time two-way thermal-loss capacity upper and lower bounds with mean thermal photon number $\bar{n}=1$. **d** Continuous-time two-way thermal-loss capacity upper and lower bounds with mean thermal photon number $\bar{n}=10$. We also show the pure-loss capacities $Q_1^{N,\mathrm{MF}}$ and $Q_2^{N,\mathrm{MF}}$ corresponding to $\bar{n}=0$ for comparison.

$$Q_2^{N,\mathrm{MF}} = \frac{\bar{g}_N}{\log(2)\sin\left[\frac{\pi}{2(N+2)}\right]}, \tag{13}$$

for one-way and two-way protocols, respectively. At large N, the continuous-time pure-loss quantum capacities saturate to the same value

$$\lim_{N\to\infty} Q_1^{N,\mathrm{MF}} = \lim_{N\to\infty} Q_2^{N,\mathrm{MF}} \equiv Q^{\max} = \frac{2\sqrt{3}}{\log(2)}g_{\max}. \tag{14}$$

The above expression represents a physical limit on the maximal achievable quantum communication rate through a transducer, $Q^{\max} \approx 5g_{\max}$ (qubit/s). The quantum communication rate through a transducer is limited by the maximal available coupling strength within the bosonic chain.

We now compare the performance of the maximally flat transducer to uniformly coupled transducers with $\forall j, \tilde{g}_j = g_{\max}$, $\tilde{\Delta}_j = -\omega_c$,

and $\tilde{\kappa}_a = \tilde{\kappa}_b = 2g_{\max}$ for even N, and $\tilde{\kappa}_a = \tilde{\kappa}_b = 2\sqrt{\frac{N+3}{N+1}}g_{\max}$ for odd N (see Methods).

The optimal efficiency functions for N-stage uniform transducers are shown in Fig. 3a and their continuous-time one-way pure-loss capacities, $Q_1^{N,\mathrm{Uni}}$, as a function of N are shown in orange in Fig. 3b. One can see that a N-stage maximally flat transducer may transmit about twice amount of quantum information per unit time compared to a N-stage uniform transducer with a uniform coupling rate $g_{\max}$. The achievable quantum communication rate can be even lower for random transducer parameters.

## Transducers under thermal noise

For realistic transduction schemes within a noisy environment, the quantum capacity will decrease due to the effect of thermal noise. The quantum capacities of Gaussian thermal-loss channels have yet to be analytically determined, but we can approach their values using additive upper and lower bound expressions. We now extend the continuous-time quantum capacity for thermal-loss channels with non-zero $\bar{n}$. In typical experimental situations, the conversion bandwidth is much smaller than the frequency scale of the thermal environment, and thus the change in the mean thermal photon number should be negligible within the conversion band. Therefore, we will treat $\bar{n}$ as a constant in evaluating the continuous-time quantum capacities. For one(two)-way scenario, we can define the continuous-time one(two)-way thermal-loss capacity lower(upper) bound for transducers as

$$Q_{1(2),\bar{n},L(U)} \equiv \int q_{1(2),\bar{n},L(U)}[\omega]d\omega/2\pi, \tag{15}$$

where $q_{1(2),\bar{n},L(U)}$ is the discrete-time one(two)-way thermal-loss capacity lower(upper) bound (see Methods).

The continuous-time quantum capacities of maximally flat transducers with different mean thermal photon numbers are shown in Fig. 4. One can see that the quantum capacities of maximally flat transducers are less susceptible to thermal loss at large N, and the difference between the upper bound, lower bound, and $Q^{\max}$ also vanishes at large N (see Methods for analytical expansions). Based on the above property and numerical evidence (see Methods), it is highly likely that maximally flat transducers are still optimal under the effect of thermal loss.

## Discussion

We have used the continuous-time quantum capacities to characterize the performance of direct quantum transducers. By considering the generic physical model of an externally connected bosonic chain with a bounded coupling rate $g_{\max}$, we showed that the maximal qubit communication rate of a transducer is given by $Q^{\max} \approx 5g_{\max}$. Such maximal capacity is achieved by maximally flat N-stage quantum transducers with $N \to \infty$. Note that our result has no contradiction to the Lieb-Robinson bound[43]—after signals arrive at a delayed time, increasing with N as predicted by Lieb and Robinson, the qubit communication rate is upper-bounded by the quantum capacity of the transducer that saturates to a finite value $Q^{\max}$ at large N in the optimal scenario.

This work provides a fundamental limit of transducer capacities in terms of coupling strength, and offers a quantitative comparison for direct transducers across platforms that consolidates distinct metrics of efficiency, bandwidth, and added thermal noise. Our method can be directly extended to transducers with intrinsic losses by considering the dependence of the conversion efficiency $\eta_N$ on the intrinsic dissipation rates[12,35]. Intriguing future works include exploring bosonic encodings, such as GKP codes[44], to approach the quantum capacity bound and investigating

superadditivity of general quantum capacities. Here we have focused on direct transducers that can be well-modeled as a Gaussian thermal-loss channel with neither amplification gain nor access to the reflective signal. A more general framework incorporating disparate transduction schemes, like direct transduction with amplification[45] due to extra two-mode squeezing couplings, or entanglement-based[46-48], adaptive-based[49], and interference-based[50,51] transductions that involve the reflective signal, is left as an open frontier to be explored.

## Methods

### Conversion efficiency of $N$-stage quantum transducers

We consider $N$-stage quantum transducers composed of a coupled bosonic chain with a Hamiltonian

$$\hat{H}_N = -\sum_{j=1}^{N+2} \Delta_j \hat{m}_j^\dagger \hat{m}_j + \sum_{j=1}^{N+1} g_j \left( \hat{m}_j^\dagger \hat{m}_{j+1} + \hat{m}_{j+1}^\dagger \hat{m}_j \right), \quad (16)$$

where $\hat{m}_j^\dagger, \hat{m}$ are the creation and annihilation operators of mode $j$, $\Delta_j$ is the detuning of mode $j$ in the rotating frame, and $g_j$ represents the coupling strength between neighboring modes. We can take $g_j$'s to be real and positive without loss of generality by absorbing their phases into mode operators. The conversion efficiency of a $N$-stage transducer without intrinsic loss is given by[35]

$$\eta_N[\omega] = \left| \frac{\sqrt{\kappa_a \kappa_b} \prod_{j=1}^{N+1} g_j}{D_N[\omega]} \right|^2, \quad (17)$$

where $D_N[\omega]$ is the determinant of a $(N+2) \times (N+2)$ tridiagonal matrix

$$D_N[\omega] \equiv \begin{vmatrix} \chi_a^{-1} & ig_1 & 0 & \cdots & \cdots & 0 \\ ig_1 & \chi_2^{-1} & ig_2 & & & \vdots \\ 0 & ig_2 & \ddots & \ddots & \ddots & \vdots \\ \vdots & \ddots & \ddots & \ddots & \ddots & 0 \\ \vdots & & & \ddots & \ddots & ig_{N+1} \\ 0 & \cdots & \cdots & 0 & ig_{N+1} & \chi_b^{-1} \end{vmatrix}. \quad (18)$$

Here $\chi_j = (i(\omega + \Delta_j) + \kappa_j/2)^{-1}$ is the susceptibility of mode $\hat{m}_j$, with $\kappa_1 = \kappa_a$, $\kappa_{N+2} = \kappa_b$, and $\kappa_j = 0$ otherwise.

### Physical parameters of maximally flat transducers

In this section we will prove that the optimal parameters given in Eqs. (8) and (9) give rise to maximally flat efficiency for transducers. Consider a $(N+2) \times (N+2)$ tridiagonal matrix $F_{N+2}$ defined as

$$F_{N+2} \equiv \begin{pmatrix} -\kappa_a^\star/2 & ig_1^\star & 0 & \cdots & 0 \\ ig_1^\star & 0 & \ddots & \ddots & \vdots \\ 0 & \ddots & \ddots & \ddots & 0 \\ \vdots & \ddots & \ddots & 0 & ig_{N+1}^\star \\ 0 & \cdots & 0 & ig_{N+1}^\star & \kappa_b^\star/2 \end{pmatrix}. \quad (19)$$

The generalized matching condition of the transducer with these parameters $\kappa_a^\star$, $\kappa_b^\star$, $\Delta_j^\star$'s, and $g_j^\star$'s is given by $M_N^\star[\omega] = \det \left( i(\omega - \omega_c) \mathbb{I}_{N+2} + F_{N+2} \right) = 0$, with the physical interpretation of generalized impedance matching criteria that leads to unity conversion efficiency and zero reflection[35].

This matrix $F_{N+2}$ is a nilpotent matrix such that all its eigenvalues are 0 and thus $M_N^\star[\omega] = (i(\omega - \omega_c))^{N+2}$, since it is a similarity transformation of another nilpotent matrix $A_{N+2}$[52] up to an energy scaling,

$$F_{N+2} = 2\sqrt{\sin\left[\frac{\pi}{2(N+2)}\right] \sin\left[\frac{3\pi}{2(N+2)}\right]} g_{\max} P_{N+2}^{-1} A_{N+2} P_{N+2}, \text{ where}$$

$$A_{N+2} \equiv \begin{pmatrix} -f_1 & f_1 & 0 & \cdots & 0 \\ -f_2 & 0 & f_2 & \ddots & \vdots \\ 0 & \ddots & \ddots & \ddots & 0 \\ \vdots & \ddots & \ddots & 0 & f_{N+1} \\ 0 & \cdots & 0 & -f_{N+2} & f_{N+2} \end{pmatrix}, \quad (20)$$

$$P_{N+2} \equiv \begin{pmatrix} 1 & 0 & \cdots & \cdots & 0 \\ 0 & i\sqrt{\frac{f_2}{f_1}} & \ddots & & \vdots \\ \vdots & \ddots & \ddots & \ddots & \vdots \\ \vdots & & \ddots & (i)^N\sqrt{\frac{f_{N+1}}{f_1}} & 0 \\ 0 & \cdots & \cdots & 0 & (i)^{N+1}\sqrt{\frac{f_{N+2}}{f_1}} \end{pmatrix}, \quad (21)$$

and $f_k = \frac{1}{2\sin\left[\frac{(2k-1)\pi}{2(N+2)}\right]}$. In other words, this choice of optimal parameters leads to a $(N+2)$-fold degenerate root at $\omega = \omega_c$ to achieve unity conversion efficiency.

For transducers without intrinsic loss, which can be modeled as lossless beam splitters, the transmittance $\eta_N[\omega]$ is related to the reflectance $R_N[\omega]$ by a simple equation $1 - \eta_N[\omega] = R_N[\omega]$. Given the expression of the reflectance

$$R_N^\star[\omega] = \frac{|M_N^\star[\omega]|^2}{|D_N^\star[\omega]|^2}, \quad (22)$$

where the superscript $\star$ denotes the association with MF parameters $\kappa_a^\star$, $\kappa_b^\star$, $\Delta_j^\star$'s, and $g_j^\star$'s, along with the $N$-stage conversion efficiency expression Eq. (17), we arrive at the maximally flat efficiency of transducers

$$\eta_N^\star[\omega] = 1 - R_N^\star[\omega] = \frac{\kappa_a^\star \kappa_b^\star \prod_{j=1}^{N+1} g_j^{\star 2}}{(\omega - \omega_c)^{2(N+2)} + \kappa_a^\star \kappa_b^\star \prod_{j=1}^{N+1} g_j^{\star 2}} = \eta_N^{\mathrm{MF}}[\omega]. \quad (23)$$

### Correspondence between maximally flat transducers and Butterworth filters

A $N$-stage transducer with maximally flat design is a direct analog to a $(N+2)$th order Butterworth low-pass electric filter[53]. The $(N+2)$th order Butterworth filter has a frequency response (gain)

$$|t_{N+2}^{\mathrm{BW}}[\omega]|^2 = \frac{1}{(\omega/\omega_{\mathrm{cut}})^{2(N+2)} + 1}, \quad (24)$$

where $t_{N+2}^{\mathrm{BW}}[\omega]$ is the transmission coefficient of the Butterworth filter with a cutoff frequency $\omega_{\mathrm{cut}}$. The frequency response of the Butterworth filter is identical to the conversion efficiency function of a maximally flat transducer while working in the rotating frame that sets the unity-efficiency conversion frequency at $\omega_c = 0$.

Moreover, a rigorous connection between the physical parameters of open-bosonic-chain transducers and electric ladder networks has been established[35]. One can verify the correspondence between a $N$-stage maximally flat transducer and a $(N+2)$th order

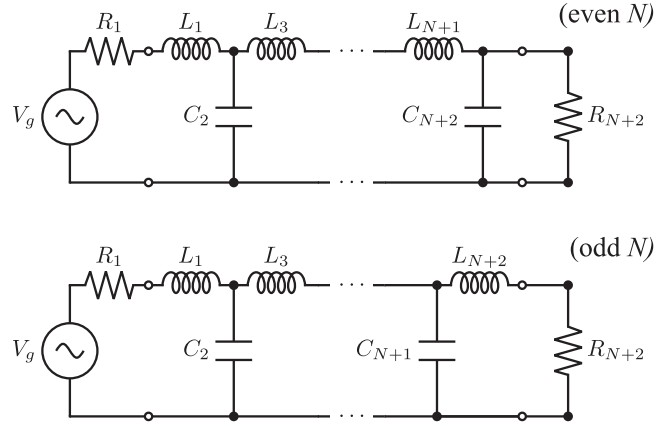

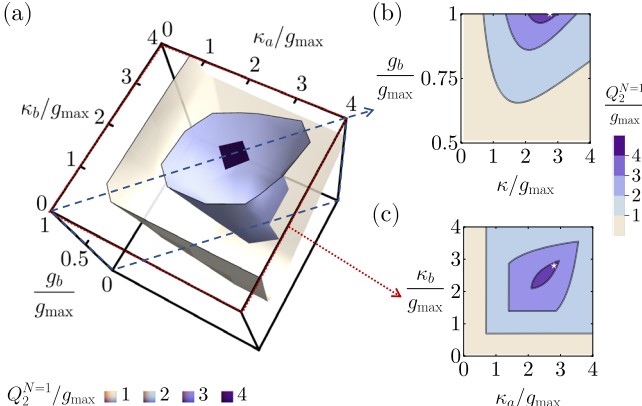

**Fig. 5 | Butterworth filter network design.** ($N$+2)th order Butterworth filter network with normalized circuit elements $R_1 = R_{N+2} = 1$, $L_j = 2\sin\left[\frac{(2j-1)\pi}{2(N+2)}\right]$, and $C_j = 2\sin\left[\frac{(2j-1)\pi}{2(N+2)}\right]$ such that $\omega_{cut} = 1$[53].

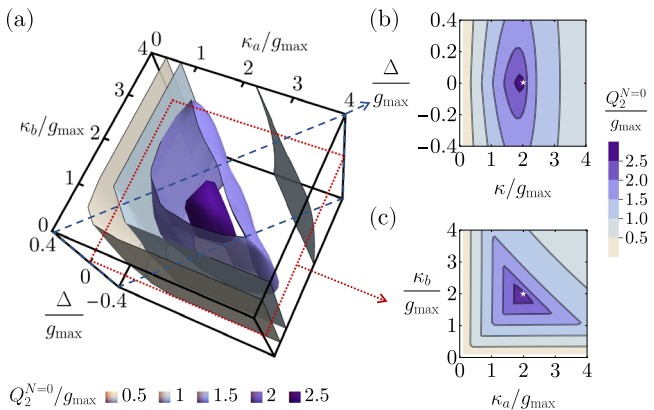

**Fig. 6 | Diagrams for the numerical search of the optimized 0-stage transducer parameters that can attain the highest possible continuous-time two-way pure-loss quantum capacity. a** Contour plot of the continuous-time two-way pure-loss capacity for $N = 0$, $Q_2^{N=0}$, in the parameter space of $\kappa_a$, $\kappa_b$, and $\Delta \equiv \Delta_a - \Delta_b$. **b** A slice in the parameter space with symmetric external coupling rates $\kappa_a = \kappa_b = \kappa$. The white star represents the location of the maximally flat parameters. **c** A slice in the parameter space under the resonant condition $\Delta = 0$. The white star represents the location of the maximally flat parameters.

Butterworth filter by showing that

$$\kappa_a^\star/\bar{g}_N/2 = R_1/L_1,$$

$$\begin{cases} g_j^{\star 2}/\bar{g}_N^2 = L_j^{-1}C_{j+1}^{-1}, & \text{odd } j \\ g_j^{\star 2}/\bar{g}_N^2 = C_j^{-1}L_{j+1}^{-1}, & \text{even } j \end{cases},$$

$$\begin{cases} \kappa_b^\star/\bar{g}_N/2 = R_{N+2}L_{N+2}^{-1}, & \text{odd } N \\ \kappa_b^\star/\bar{g}_N/2 = R_{N+2}^{-1}C_{N+2}^{-1}, & \text{even } N \end{cases}, \qquad (25)$$

where $R_j$, $L_j$, and $C_j$ correspond to resistance, inductance, and capacitance of the normalized Butterworth filter as shown in Fig. 5. One may also add generalized resistances $\mathcal{R}_j$'s of imaginary values to include the shifts in the detunings, $\Delta_j = -\omega_c$. The nilpotent matrix argument provided in the previous section can also serve as a mathematical proof for the analytical formulas of Butterworth filter circuit parameters, which were originally determined from observation[53].

## Numerical evidence for the optimality of maximally flat transducers

In this section, we provide numerical evidence showing that for $N$-stage direct transduction, under the physical constraint

**Fig. 7 | Diagrams for the numerical search of the optimized 1-stage parameters to achieve the highest possible continuous-time two-way pure-loss capacity under the resonant assumption $\Delta_a = \Delta_2 = \Delta_b$. a** Contour plot of the continuous-time two-way pure-loss capacity for $N = 1$, $Q_2^{N=1}$, in the parameter space of $\kappa_a$, $\kappa_b$, and $g_b$, assuming $g_a = g_{max}$. **b** A slice in the parameter space with a symmetric external coupling rate $\kappa_a = \kappa_b = \kappa$. The white star represents the point at the maximally flat parameters. **c** A slice in the parameter space with the saturated coupling condition $g_b = g_{max}$. The white star represents the point at the maximally flat parameters.

$\forall j, g_j \leq g_{max}$, the set of parameters for a maximally flat transducer likely gives rise to global maxima of the continuous-time pure-loss quantum capacities $Q_1$ and $Q_2$. For the 0-stage case, we numerically optimize the continuous-time one- and two-way pure-loss quantum capacities by an exhaustive search over all the free parameters $\kappa_a$, $\kappa_b$, and $\Delta \equiv \Delta_a - \Delta_b$ in the unit of $g_{max} = g_a$. In Fig. 6a, we show a three-dimensional contour plot of the two-way continuous-time pure-loss quantum capacity for 0-stage transducers, $Q_2^{N=0}$, in the parameter space of $\kappa_a$, $\kappa_b$, and $\Delta$. To identify the optimal parameters, we show the two slices in the parameter space where the maximum locates. A 2D slice assuming symmetric external couplings $\kappa_a = \kappa_b = \kappa$ is presented in Fig. 6b, and another slice under the resonant condition between the two modes $\Delta = 0$ is shown in Fig. 6c. We can see that the set of analytically determined maximally flat parameters, $\Delta_a^\star = \Delta_b^\star (= -\omega_c)$ and $\kappa_a^\star = \kappa_b^\star = 2g_{max}$ as marked by the white star, coincides with the location of the numerical maximum. The same finding applies to the continuous-time one-way pure-loss quantum capacity, which has a qualitatively similar structure in the parameter space.

For the 1-stage case, we numerically optimize the two-way continuous-time quantum capacity by an exhaustive search over five free parameters $\kappa_a$, $\kappa_b$, $\Delta_b' \equiv \Delta_a - \Delta_b$, $\Delta_2' \equiv \Delta_a - \Delta_2$, and $g_b$, in the unit of $g_{max} = g_a$. We find that the global maximum is achieved when the three modes are resonant, $\Delta_a = \Delta_2 = \Delta_b$. Under the all-resonant assumption, we present the numerical search over the rest of the three parameters $\kappa_a$, $\kappa_b$, and $g_b$ in Fig. 7. In Fig. 7a, we show a three-dimensional contour plot of the continuous-time two-way pure-loss quantum capacity for 1-stage transducers, $Q_2^{N=1}$, in the parameter space of $\kappa_a$, $\kappa_b$, and $g_b$. To identify the optimal parameters, we again show two slices in the parameter space where the maximum locates. A 2D slice assuming symmetric external couplings $\kappa_a = \kappa_b = \kappa$ is presented in Fig. 7b, and another slice with symmetric internal couplings $g_b = g_a = g_{max}$ is shown in Fig. 7c. We can see that the set of the analytically-determined maximally flat parameters, $\Delta_a^\star = \Delta_2^\star = \Delta_b^\star (= -\omega_c)$, $\kappa_a^\star = \kappa_b^\star = 2\sqrt{2}g_{max}$, and $g_a^\star = g_b^\star = g_{max}$ as indicated by the white star, coincides with the location of the numerical maximum.

For higher number of stages, we assume the system is under the all-resonant condition and is symmetric, $\forall j, \Delta_j = -\omega_c$, $\kappa_a = \kappa_b$, and $g_j = g_{N+2-j}$, to reduce the number of optimization parameters. For the continuous-time one- and two-way pure-loss quantum

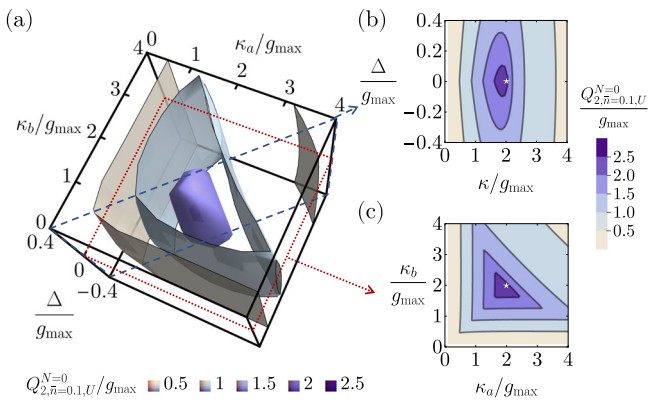

**Fig. 8 | Diagrams for the numerical search of the optimized 0-stage transducer parameters that can attain the highest possible continuous-time two-way thermal-loss quantum capacity upper bound with a mean thermal photon number $\bar{n} = 0.1$. a** Contour plot of the continuous-time two-way thermal-loss capacity upper bound for $N = 0$, $Q_{2,\bar{n}=0.1,U}^{N=0}$, in the parameter space of $\kappa_a$, $\kappa_b$, and $\Delta \equiv \Delta_a - \Delta_b$. **b** A slice in the parameter space with symmetric external coupling rates $\kappa_a = \kappa_b = \kappa$. The white star represents the location of the maximally flat parameters. **c** A slice in the parameter space under the resonant condition $\Delta = 0$. The white star represents the location of the maximally flat parameters.

capacities, based upon the above conjectures observed from the 0- and 1-stage cases, we have numerically verified the global optimality of the maximally flat transducers up to $N=5$. Our findings suggest a strong numerical evidence that the maximally flat transducers are highly likely the optimal choices to achieve globally maximal quantum capacities at any given number of intermediate stage $N$.

### Uniform coupling transducers

Here we discuss the optimized parameters for uniformly coupled transducers, $\forall j, \tilde{g}_j = g_{\max}$. After numerical optimizing over $\Delta_j$, $\kappa_a$, and $\kappa_b$ in search of maximal $Q_1$ and $Q_2$, we find that optimal designs of uniform transducers also show features of flatness around the ideal conversion frequency $\omega_c$ such that

$$
\begin{aligned}
\frac{\partial \eta_N^{\text{Uni}}[\omega]}{\partial \omega}\Big|_{\omega=\omega_c} &= \cdots = \frac{\partial^3 \eta_N^{\text{Uni}}[\omega]}{\partial \omega^3}\Big|_{\omega=\omega_c} = 0, N \text{ even}, \\
\frac{\partial \eta_N^{\text{Uni}}[\omega]}{\partial \omega}\Big|_{\omega=\omega_c} &= \cdots = \frac{\partial^5 \eta_N^{\text{Uni}}[\omega]}{\partial \omega^5}\Big|_{\omega=\omega_c} = 0, N \text{ odd}.
\end{aligned} \quad (26)
$$

The corresponding optimized parameters denoted by tilde are $\forall j, \tilde{\Delta}_j = -\omega_c$, $\tilde{\kappa}_a = \tilde{\kappa}_b = 2g_{\max}$ for even $N$, and $\tilde{\kappa}_a = \tilde{\kappa}_b = 2\sqrt{\frac{N+3}{N+1}}g_{\max}$ for odd $N$. The global optimality of these parameters has been numerically verified up to $N=10$ under the symmetric assumption $\kappa_a = \kappa_b$ and the resonant condition $\forall j, \Delta_j = -\omega_c$.

From Fig. 3, we observe that optimal uniform transducers with odd $N$ have higher $Q_1$ than those with even $N$, which may be explained by the two extra orders of flatness around $\omega_c$ and that odd transducers have stronger coupling rates to the external ports.

### Bounds on the discrete-time quantum capacities of thermal-loss channels

To our knowledge, the tightest lower bound on the discrete-time one-way thermal-loss quantum channel capacity is[38]

$$
q_{1,\bar{n},L}[\omega] = \max\left\{\log_2\left[\frac{\eta[\omega]}{1-\eta[\omega]}\right] - h(\bar{n}[\omega]), 0\right\}, \quad (27)
$$

$$
h(x) \equiv (x+1)\log_2(x+1) - x\log_2 x. \quad (28)
$$

For a $N$-stage maximally flat transducer, we can find an analytical expression for its continuous-time thermal-loss quantum capacity lower bound,

$$
\begin{aligned}
Q_{1,\bar{n},L}^{N,\text{MF}} &= \frac{2(N+2)}{\pi\log(2)}\left[\left(1+\frac{1}{\bar{n}}\right)^{\bar{n}}(1+\bar{n})\right]^{-\frac{1}{2(N+2)}}\bar{g}_N \\
&\approx \left[\frac{2(N+2)}{\pi\log(2)} - \frac{(1-\log(\bar{n}))\bar{n}}{\pi\log(2)}\right]\bar{g}_N + \mathcal{O}(\bar{n}^2) \\
&\approx Q^{\max} - \frac{\sqrt{3}[1-\log(\bar{n})]\bar{n}}{N\log(2)}g_{\max} + \mathcal{O}\left(\frac{1}{N^2}\right),
\end{aligned} \quad (29)
$$

where we have expanded $Q_{1,\bar{n},L}$ around small thermal-photon number $\bar{n} \approx 0$ in the second line, and then further expand the expression around large $N$ in the last approximation.

On the other hand, there is no single analytical form for the tightest upper bound on the discrete-time one-way thermal-loss capacity. Here we combine the three best upper bound formulas known and define $q_{1,\bar{n},U}[\omega]$ as

$$
q_{1,\bar{n},U}[\omega] = \min\left\{q_{1,\bar{n},U,\text{twist}}[\omega], q_{1,\bar{n},U,\text{DE}}[\omega], q_{2,\bar{n},U}[\omega]\right\}. \quad (30)
$$

Here $q_{1,\bar{n},\text{twist}}$ is the upper bound attained by a twisted version of a quantum-limited attenuator and amplifier decomposition of thermal attenuators[54,55],

$$
q_{1,\bar{n},\text{twist}}[\omega] = \max\left\{\log_2\left[\frac{\eta[\omega]-(1-\eta[\omega])\bar{n}[\omega]}{(1-\eta[\omega])(\bar{n}[\omega]+1)}\right], 0\right\}, \quad (31)
$$

$q_{1,\bar{n},\text{DE}}$ is the upper bound given by the degradable extensions of thermal-loss channels[56],

$$
q_{1,\bar{n},\text{DE}}[\omega] = \max\left\{\log_2\left[\frac{\eta[\omega]}{1-\eta[\omega]}\right] + h((1-\eta[\omega])\bar{n}[\omega]) - h(\eta[\omega]\bar{n}[\omega]), 0\right\}, \quad (32)
$$

and $q_{2,\bar{n},U}$ is the upper bound on the quantum capacity of thermal-loss channels assisted by two-way classical communication and local operations[40],

$$
q_{2,\bar{n},U}[\omega] = \max\left\{-\log_2\left[(1-\eta[\omega])\eta[\omega]^{\bar{n}[\omega]}\right] - h(\bar{n}[\omega]), 0\right\}. \quad (33)
$$

These three formulas above give rise to the tightest upper-bound values in different parameter regimes, and thus we combine all three of them to achieve the best upper bound formula.

For two-way protocols, the best known discrete time two-way thermal-loss capacity lower bound is[40]

$$
q_{2,\bar{n},L}[\omega] = \max\left\{-\log_2[1-\eta[\omega]] - h(\bar{n}[\omega]), 0\right\}, \quad (34)
$$

and we calculate the analytical formula for the continuous-time two-way thermal-loss capacity lower bound of a $N$-stage maximally flat transducer as

$$
\begin{aligned}
Q_{2,\bar{n},L}^{\text{MF}} &= \frac{2(N+2)}{\pi\log(2)k(\bar{n})^{\frac{1}{2(N+2)}}} {}_2F_1\left[1, \frac{1}{2(N+2)}, 1+\frac{1}{2(N+2)}, -\frac{1}{k(\bar{n})}\right]\bar{g}_N \\
&\approx \left\{\frac{1}{\log(2)}\csc\left[\frac{\pi}{2(N+2)}\right] - \frac{2(N+2)(\bar{n}-\bar{n}\log(\bar{n}))^{\frac{2N+3}{2(N+2)}}}{\pi(2N+3)\log(2)}\right\}\bar{g}_N + \mathcal{O}(\bar{n}^2) \\
&\approx Q^{\max} - \frac{\sqrt{3}[1-\log(\bar{n})]\bar{n}}{N\log(2)}g_{\max} + \mathcal{O}\left(\frac{1}{N^2}\right),
\end{aligned} \quad (35)
$$

$$
k(x) \equiv (1+x)(1+x^{-1})^x - 1. \quad (36)
$$

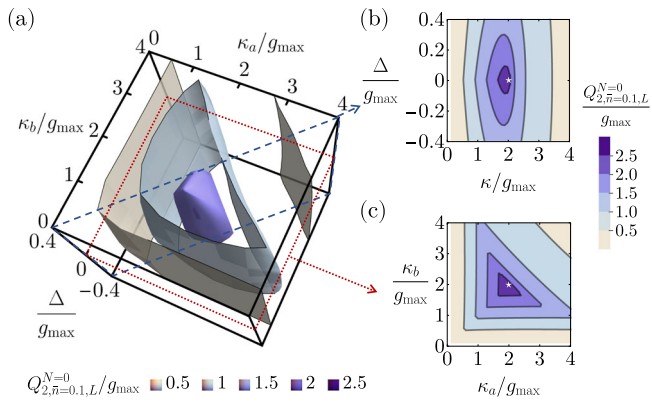

**Fig. 9 | Diagrams for the numerical search of the optimized 0-stage transducer parameters that can attain the highest possible continuous-time two-way thermal-loss quantum capacity lower bound with a mean thermal photon number $\bar{n}$ = 0.1. a** Contour plot of the continuous-time two-way thermal-loss capacity lower bound for $N = 0$, $Q_{2,\bar{n}=0.1,L}^{N=0}$, in the parameter space of $\kappa_a$, $\kappa_b$, and $\Delta \equiv \Delta_a - \Delta_b$. **b** A slice in the parameter space with symmetric external coupling rates $\kappa_a = \kappa_b = \kappa$. The white star represents the location of the maximally flat parameters. **c** A slice in the parameter space under the resonant condition $\Delta = 0$. The white star represents the location of the maximally flat parameters.

Here $_2F_1$ is the hypergeometric function.

For a maximally flat $N$-stage transducer, its continuous-time two-way thermal-loss capacity upper bound associated with $q_{2,\bar{n},U}[\omega]$[40] is

$$
\begin{aligned}
Q_{2,\bar{n},U}^{N,MF} &= \frac{2(N+2)}{\pi \log(2)\bar{n}^{\frac{1}{2(N+2)}}} \left\{ (\bar{n}+1)_2F_1\left[1, \frac{1}{2(N+2)}, 1+\frac{1}{2(N+2)}, -\frac{1}{\bar{n}}\right] - \bar{n} \right\} \bar{g}_N \\
&\approx \left[ \frac{(1+\bar{n})}{\log(2)} \csc\left[\frac{\pi}{2(N+2)}\right] - \frac{4(N+2)^2 \bar{n}^{\frac{2N+3}{2(N+2)}}}{\pi(2N+3)\log(2)} \right] \bar{g}_N + \mathcal{O}(\bar{n}^2) \\
&\approx Q^{max} - \frac{\sqrt{3}[1-\log(\bar{n})]\bar{n}}{N\log(2)} g_{max} + \mathcal{O}\left(\frac{1}{N^2}\right).
\end{aligned}
\tag{37}
$$

We have seen numerical evidence showing that maximally flat transducers are still optimal under the effect of thermal loss. In Figs. 8 and 9, we plot upper and lower bound diagrams for the numerical search of the optimal 0-stage transducer under thermal loss. Those diagrams behave qualitatively similar to the pure-loss quantum capacities in Fig. 6, and the location of the numerical maximum again coincides with the parameters of the 0-stage maximally flat transducer.

## Code availability

Source codes of the figures presented in this article are available from the corresponding author upon request.

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

## Acknowledgements

We thank Aashish Clerk, Yat Wong, Mengzhen Zhang, and Changchun Zhong for helpful discussions. We acknowledge support from the ARO (W911NF-18-1-0020, W911NF-18-1-0212), ARO MURI (W911NF-16-1-0349, W911NF-21-1-0325), AFOSR MURI (FA9550-19-1-0399, FA9550-21-1-0209), AFRL (FA8649-21-P-0781), DoE Q-NEXT, NSF (OMA-1936118, EEC-1941583, OMA-2137642), NTT Research, and the Packard Foundation (2020-71479).

## Author contributions

C.-H.W. developed the theoretical formalism, carried out the analytical calculations, and generated the figures. C.-H.W. and F.L. performed the numerical simulations. L.J. conceived the original idea and supervised the work. C.-H.W., F.L., and L.J. wrote the paper.

## Competing interests

The authors declare no competing interests.
