## [Peer Review File · Nature Communications]

Quantum Capacities of TransducersREVIEWER COMMENTS

Reviewer #1 (Remarks to the Author):

The authors considered quantum transducers, which are devices that allow to convert quantum signals between different physical systems or mediums. They modeled transducers by a series of lossy channels and calculated limits on their capacities for a particular case of maximally flat transducers. They give convincing numerical arguments that this is the optimal scenario. Finally, they present results for lower and upper bounds on capacities of maximally flat transducers assuming additional thermal noise. In my opinion the manuscript is written quite clearly and is valid, however, I found some issues that should be solved before I can recommend it for publication.

My first issue is about the model that authors consider, which is a bosonic thermal-loss channel. The choice of such a model is not motivated in the manuscript. I think authors should give such motivation and some reasons why this model is relevant and how general it is somewhere in the introduction.

In equations (2)-(5) authors write about one and two-way quantum capacities of lossy channels and their related definitions for transducers. These formulas, however, assume infinite signal energy which in practice never occurs. I think this issue should be mentioned in the discussion of these four equations.

Authors give convincing numerical evidence to why the maximally flat transducers are optimal for lossy channels. However, similar evidence for thermal noise is lacking. I think it would be good to see at least a single plot analogous to Fig. 6 or 7 of the manuscript but calculated for the thermal noise. Is there some other argument for the optimality here or maybe maximally flat transducers are not optimal in this scenario? In such a case it would be good to have a comment on this in the manuscript.

Authors establish a connection between maximally flat transducers for the loss-only model and Butterworth filters. Is there any similar connection for the thermal case as well?

Finally, a minor issue is that function F_{-1}^2 in equations (32) and (34) is not described anywhere. I suppose this is a hypergeometric function ${}_2F_1$ but this is unclear from the notation if this is so or this is a square of some other function. I recommend changing the notation to ${}_2F_1$ and writing explicitly what this symbol represents somewhere around eq. (32).

In conclusion I think the manuscript is interesting but there are some issues that need to be taken into account before I can recommend it for publication.

Reviewer #2 (Remarks to the Author):

In the manuscript "Quantum Capacities of Transducers" by C.-H. Wang, F. Li and L. Jiang the authors introduce the continuous-time pure-loss one-way and two-way quantum capacities as a metric to benchmark different types of quantum transducers. This is done by considering transducers as bosonic thermal-loss channels and using the known concept of quantum channel capacities. The continuous-time quantum capacities combine two other figures of merits, namely the efficiency and the bandwidth of a transducer, into one metric that only depends on the physical parameters of the transducer's underlying implementation.

Modeling the direct quantum transduction as a chain of coupled bosonic modes, and assuming the absence of internal losses, and under the physical constraint of bounded coupling rates the authors find that the quantum capacities are maximized for a specific choice of physical parameters of an N-stage quantum transducer. This optimized set of parameters yields a maximally flat spectral efficiency with a plateau around the maximum value of the transduction efficiency $\eta(\omega_c) = 1$ and results in the highest quantum capacity of $Q = g_{\max} 4\sqrt{3} \pi / \log(2)$.

The authors complete their analysis by providing the lower and upper bound expressions for the continuous-time quantum capacities in the presence of thermal noise.

By introducing continuous-time pure-loss quantum capacities as a new metric that can incorporate up to three different figures of merit – efficiency, bandwidth and the number of noise photons – this work enables a better way for comparison of different physical implementations of a quantum transducer and hence will have a significant impact on the field of quantum transduction. However, there are some points in the current manuscript that should be addressed before I can recommend this work for publication. The critical points are as follows:

1. On page 4 after the Eq. (6) the authors claim that they “will take g_j 's to be real and positive without loss of generality”. Could the authors comment on why this is possible in their case?
2. In the Methods section the authors justify the choice of optimal parameter set by arguing that the tridiagonal matrix $F_{(N+2)}$ is nilpotent for this particular choice of parameters and they cite a paper A. Behn et al. as reference for this. However, if I understand it correctly, this reference shows that another type of tridiagonal matrix, namely a real matrix and with a specific sign pattern is nilpotent but not a matrix of the type $F_{(N+2)}$. So I was wondering how could the authors conclude that the matrix $F_{(N+2)}$ is nilpotent and how exactly were the optimal parameter values, Eqs. (8) and (9), determined.
3. On page 6 the authors claim that “the achievable quantum communication rate is even lower for arbitrary transducer parameters without flatness feature around $\eta[\omega_c]=1$.” This statement lacks any supporting arguments. What do the authors mean when they say without flatness feature? Since there are certainly parameters, see for example the numerical study in the Methods section, that will give a value of quantum capacity that will be higher than Q^{Uni} but lower than Q^{MF} .

Some minor things:

4. In the line after Eq. (16) shouldn't it read $(N+2) \times (N+2)$ matrix instead of $(N-2) \times (N-2)$?
5. Same in the line above Eq. (18)?
6. In Eq. (32), the function F_{-1}^2 is not defined.

In summary, I think that this work will benefit the transducer community and can recommend it for publishing after my concerns are addressed in a satisfactory manner.

Response to the reviewers' comments

Chiao-Hsuan Wang, Fangxin Li, and Liang Jiang

Reply: We thank the reviewers for their time and their constructive comments. We have responded to the concerns of the reviewers below and revised our manuscript accordingly with changes highlighted in blue text. In addition, we have updated the affiliations of the corresponding author.

Reviewer #1 (Remarks to the Author)

“The authors considered quantum transducers, which are devices that allow to convert quantum signals between different physical systems or mediums. They modeled transducers by a series of lossy channels and calculated limits on their capacities for a particular case of maximally flat transducers. They give convincing numerical arguments that this is the optimal scenario. Finally, they present results for lower and upper bounds on capacities of maximally flat transducers assuming additional thermal noise. In my opinion the manuscript is written quite clearly and is valid, however, I found some issues that should be solved before I can recommend it for publication.”

My first issue is about the model that authors consider, which is a bosonic thermal-loss channel. The choice of such a model is not motivated in the manuscript. I think authors should give such motivation and some reasons why this model is relevant and how general it is somewhere in the introduction.”

Reply: To motivate why bosonic thermal-loss channel is relevant, we have added in the introduction “After sending an input signal through a quantum transducer without amplification, the output signal will be a mixture of the input signal and environmental noise. Assuming the environmental noise is thermal, the action of the transducer can be described as a bosonic thermal-loss channel that attenuates the input state and combines it with a noisy thermal state.”

“In equations (2)-(5) authors write about one and two-way quantum capacities of lossy channels and their related definitions for transducers. These formulas, however, assume infinite signal energy which in practice never occurs. I think this issue should be mentioned in the discussion of these four equations.”

Reply: We have added discussions under Eq. (5) : ” To characterize these maximal achievable rates, we have assumed that infinite energy is available at the transducers. In practice, quantum capacities of transducers shall be lower in energy-constrained scenarios [40,41]”

“Authors give convincing numerical evidence to why the maximally flat transducers are optimal for lossy channels. However, similar evidence for thermal noise is lacking. I think it would be good to see at least a single plot analogous to Fig. 6 or 7 of the manuscript but calculated for the thermal noise. Is there some other argument for the optimality here or maybe maximally flat transducers are not optimal in this scenario? In such a case it would be good to have a comment on this in the manuscript.

Authors establish a connection between maximally flat transducers for the loss-only model and Butterworth filters. Is there any similar connection for the thermal case as well?”

Reply: We thank the referee for this suggestion. We have included new figures, Fig.8 and Fig.9, (analogous to Fig.6) for numerical search of optimal parameters for thermal loss channels. Based on numerical evidence and the fact that the thermal-loss capacities lower and upper bounds will approach Q^{\max} at large N , we believe that maximally flat transducers are still optimal under the effect of thermal

loss. This optimal scenario is thus again analogous to the Butterworth filter design. We have also expanded the paragraph above Fig. 4 and added another paragraph under Fig. 8 to discuss the optimality of maximally flat transducers under thermal loss.

“Finally, a minor issue is that function F_1^2 in equations (32) and (34) is not described anywhere. I suppose this is a hypergeometric function ${}_2F_1$ but this is unclear from the notation if this is so or this is a square of some other function. I recommend changing the notation to ${}_2F_1$ and writing explicitly what this symbol represents somewhere around eq. (32).”

Reply: We thank the referee for pointing this out. Following the recommendation, we have changed the notation and written explicitly that “Here ${}_2F_1$ is the hypergeometric function” below Eq. (36).

“In conclusion I think the manuscript is interesting but there are some issues that need to be taken into account before I can recommend it for publication.”

Reviewer #2 (Remarks to the Author)

“In the manuscript “Quantum Capacities of Transducers” by C.-H. Wang, F. Li and L. Jiang the authors introduce the continuous-time pure-loss one-way and two-way quantum capacities as a metric to benchmark different types of quantum transducers. This is done by considering transducers as bosonic thermal-loss channels and using the known concept of quantum channel capacities. The continuous-time quantum capacities combine two other figures of merits, namely the efficiency and the bandwidth of a transducer, into one metric that only depends on the physical parameters of the transducer’s underlying implementation.

Modeling the direct quantum transduction as a chain of coupled bosonic modes, and assuming the absence of internal losses, and under the physical constraint of bounded coupling rates the authors find that the quantum capacities are maximized for a specific choice of physical parameters of an N-stage quantum transducer. This optimized set of parameters yields a maximally flat spectral efficiency with a plateau around the maximum value of the transduction efficiency $\eta(\omega_c)=1$ and results in the highest quantum capacity of $Q=g_{\max} \sqrt{3} \pi/\log(2)$. The authors complete their analysis by providing the lower and upper bound expressions for the continuous-time quantum capacities in the presence of thermal noise.

By introducing continuous-time pure-loss quantum capacities as a new metric that can incorporate up to three different figures of merit – efficiency, bandwidth and the number of noise photons – this work enables a better way for comparison of different physical implementations of a quantum transducer and hence will have a significant impact on the field of quantum transduction. However, there are some points in the current manuscript that should be addressed before I can recommend this work for publication. The critical points are as follows:”

“1. On page 4 after the Eq. (6) the authors claim that they “will take g_j ’s to be real and positive without loss of generality”. Could the authors comment on why this is possible in their case?”

Reply: This simplification is possible because one can absorb the phases of g_j ’s into the definition of mode creation and annihilation operators such that the relevant parameter for our model is $|g_j|$. To clarify

this point, we have added the explicit form of Hamiltonian Eq. (16) and provide explanations below the equation. One can also see this by looking at the conversion efficiency expression Eq. (17) which only depends on $|g_j's|^2$ and thus the phase of $g_j's$ does not change the physics of the system.

“2. In the Methods section the authors justify the choice of optimal parameter set by arguing that the tridiagonal matrix $F_{(N+2)}$ is nilpotent for this particular choice of parameters and they cite a paper A. Behn et al. as reference for this. However, if I understand it correctly, this reference shows that another type of tridiagonal matrix, namely a real matrix and with a specific sign pattern is nilpotent but not a matrix of the type $F_{(N+2)}$. So I was wondering how could the authors conclude that the matrix $F_{(N+2)}$ is nilpotent and how exactly were the optimal parameter values, Eqs. (8) and (9), determined.”

Reply: On page 9, second paragraph, we have added the explicit formula of the similarity transformation between F_{N+2} and the specific patterned tridiagonal matrix A_{N+2} from the reference. The optimal parameter values were determined by first observing the correspondence between maximally flat transducers and the Butterworth filters, and then later proved by the nilpotent matrix argument.

“3. On page 6 the authors claim that “the achievable quantum communication rate is even lower for arbitrary transducer parameters without flatness feature around $\eta[\omega_c]=1$.” This statement lacks any supporting arguments. What do the authors mean when they say without flatness feature? Since there are certainly parameters, see for example the numerical study in the Methods section, that will give a value of quantum capacity that will be higher than Q^{Uni} but lower than Q^{MF} .”

Reply: We were meant to say that transducers with random-shaped efficiency functions typically have even lower quantum capacity. To avoid confusion, we have updated the sentence as “The achievable quantum communication rate can be even lower for random transducer parameters.”

“Some minor things:

4. In the line after Eq. (16) shouldn't it read $(N+2) \times (N+2)$ matrix instead of $(N-2) \times (N-2)$?

5. Same in the line above Eq. (18)?”

Reply: We thank the reviewer for pointing this out. We have corrected the above typos.

“6. In Eq. (32), the function F_{-1}^2 is not defined.”

Reply: We have now added a sentence “Here ${}_2F_1$ is the hypergeometric function” below Eq. (36).

“In summary, I think that this work will benefit the transducer community and can recommend it for publishing after my concerns are addressed in a satisfactory manner.”

REVIEWERS' COMMENTS

Reviewer #1 (Remarks to the Author):

The authors have answered all my comments and I can now recommend this manuscript for publication.

Reviewer #2 (Remarks to the Author):

The authors satisfactorily addressed my previous concern and I recommend the manuscript for publication.